# Flow-Based Feature Fusion for Vehicle-Infrastructure Cooperative 3D Object Detection

**Haibao Yu**[1,2], **Yingjuan Tang**[2,3], **Enze Xie**[1], **Jilei Mao**[2], **Ping Luo**[1,4], **Zaiqing Nie**[2] *
[1]The University of Hong Kong
[2]Institute for AI Industry Research (AIR), Tsinghua University
[3]Beijing Institute of Technology  [4]Shanghai AI Laboratory

## Abstract

Cooperatively utilizing both ego-vehicle and infrastructure sensor data can significantly enhance autonomous driving perception abilities. However, the uncertain temporal asynchrony and limited communication conditions can lead to fusion misalignment and constrain the exploitation of infrastructure data. To address these issues in vehicle-infrastructure cooperative 3D (VIC3D) object detection, we propose the Feature Flow Net (FFNet), a novel cooperative detection framework. FFNet is a flow-based feature fusion framework that uses a feature flow prediction module to predict future features and compensate for asynchrony. Instead of transmitting feature maps extracted from still-images, FFNet transmits feature flow, leveraging the temporal coherence of sequential infrastructure frames. Furthermore, we introduce a self-supervised training approach that enables FFNet to generate feature flow with feature prediction ability from raw infrastructure sequences. Experimental results demonstrate that our proposed method outperforms existing cooperative detection methods while only requiring about 1/100 of the transmission cost of raw data and covers all latency in one model on the DAIR-V2X dataset. The code is available at https://github.com/haibao-yu/FFNet-VIC3D.

## 1    Introduction

Accurate 3D object detection is a critical task in autonomous driving as it provides crucial information about the location and classification of surrounding obstacles. Traditional 3D object detection methods rely on onboard sensor data from the ego vehicle, which has a limited perception field and often fails in blind or long-range zones, resulting in safety concerns. To address these challenges, vehicle-infrastructure cooperative autonomous driving has gained much attention, particularly using infrastructure sensors like cameras and LiDARs, which are usually installed higher than ego vehicles, providing a broader field of view (40; 42; 25; 24). By utilizing additional infrastructure sensor data, it is possible to obtain more meaningful information and improve autonomous driving perception ability. In this paper, we focus on solving the vehicle-infrastructure cooperative 3D (VIC3D) object detection problem to enhance the safety and performance of autonomous driving systems in challenging traffic scenarios.

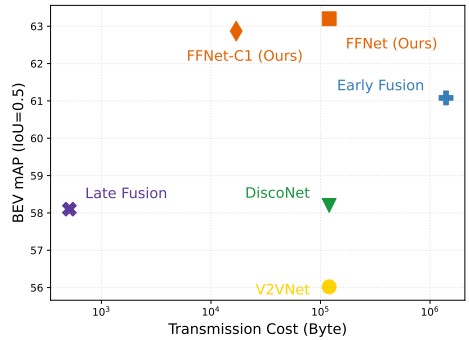

Figure 1: **Performance** $vs.$ **Transmission Cost on DAIR-V2X Dataset.** All results are reported with $200ms$ latency. FFNet achieves a new state-of-the-art 62.87% mAP@BEV while only requiring about 1/100 of the transmission cost of early fusion.

---

*Corresponding author. Work done while at AIR.

37th Conference on Neural Information Processing Systems (NeurIPS 2023).

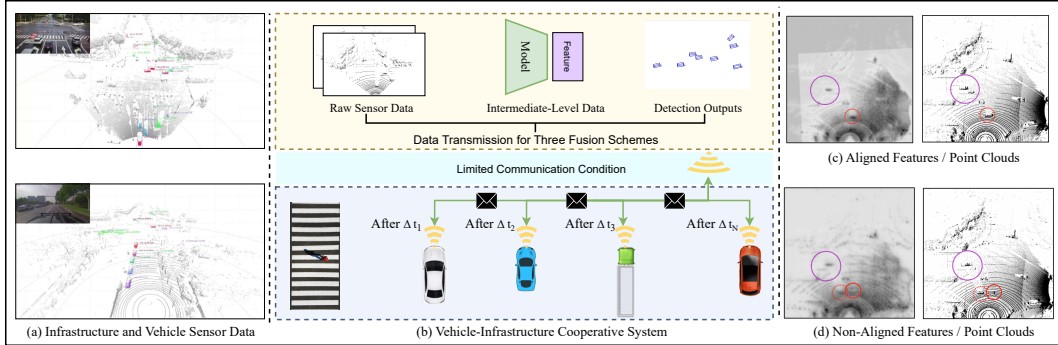

Figure 2: Vehicle-Infrastructure Cooperative 3D Object Detection. (a) Infrastructure $vs$ Vehicle Sensor Data. Infrastructure sensor data can provide abundant information for autonomous driving with a broader perception field compared to vehicle sensor data. (b) Vehicle-Infrastructure Cooperative System. There are three potential data forms for transmission: raw data for early fusion, intermediate-level data for middle fusion, and detection outputs for late fusion. Due to limited communication conditions, the infrastructure information may be received by any vehicle with an uncertain latency, resulting in uncertain temporal asynchrony. (c-d) Aligned and Non-aligned Point Clouds and Features. Non-aligned point clouds and features can cause fusion misalignment and affect the exploitation of infrastructure data, potentially impacting the performance of the cooperative detection.

The VIC3D problem can be formulated as a multi-sensor detection problem under constrained communication bandwidth, presenting two main challenges. First, infrastructure data can be received by any vehicle, and the data captured by ego-vehicle sensors and received from infrastructure devices have asynchronous timestamps with uncertain differences. Second, the communication bandwidth between the two-side devices is limited. Recent studies (40; 14; 33) have attempted to address this problem and proposed three major fusion frameworks for cooperative detection: early fusion, late fusion, and middle fusion. Early fusion involves transmitting raw data like raw point clouds, while late fusion uses detection outputs for object-level fusion. Middle fusion utilizes intermediate-level features for feature fusion, striking a balance between preserving valuable information and reducing redundant transmission. However, existing middle-fusion solutions (15; 14; 33) overlook the challenge of temporal asynchrony explicitly, leading to fusion misalignment that affects detection results, as depicted in Figure 2. **This paper aims to address these challenges in a simple and unified manner. Specifically, we propose the Feature Flow Net (FFNet), a novel cooperative detection framework that simultaneously overcomes the issues of uncertain temporal asynchrony and communication bandwidth limitations in VIC3D object detection.**

As depicted in Figure 3, FFNet comprises several steps, including generating feature flow from sequential infrastructure frames, transmitting the compressed feature flow, and fusing it with ego-vehicle features to obtain detection output. The feature flow is a critical component of FFNet, serving as a feature prediction function that enables alignment with ego-vehicle features and eliminates fusion errors arising from temporal asynchrony. To reduce transmission costs while preserving valuable information and temporal prediction ability, we employ attention masks and quantization methods to further compress the feature flow before transmission. Furthermore, we introduce a self-supervised approach to train the feature flow generator. This approach involves constructing ground truth features using raw infrastructure sequences, eliminating the need for manual labeling. The feature flow captures rich temporal correlations extracted from the raw infrastructure sequence and exhibits the ability to predict infrastructure features at any future time, making it well-suited for addressing the challenge of uncertain temporal asynchrony in VIC3D object detection. To the best of our knowledge, this is the first time feature flow has been utilized in multi-sensor object detection to address the issue of temporal misalignment in intermediate levels.

We implemented the proposed FFNet framework on the DAIR-V2X dataset (40), which consists of real-world driving scenarios in challenging traffic intersections. To demonstrate the effectiveness of FFNet, we conducted performance comparisons with several existing cooperative detection methods, including V2VNet (30) and DiscoNet (19). The experimental results reveal that FFNet surpasses all other cooperative methods while utilizing only about 1/100 of the transmission cost required for transmitting raw data. Furthermore, our method effectively addresses the challenge of temporal asynchrony and overcoming latency variations ranging from $100ms$ to $500ms$ in one model. Experiments encompassing additional V2V (vehicle-to-vehicle) scenarios will soon be public.

The main contributions of this work are as follows:

- We propose Feature Flow Net (FFNet), a flow-based feature fusion framework for VIC3D object detection. FFNet transmits feature flow to generate aligned features for data fusion, providing a simple and unified manner to transmit valuable information for fusion while addressing the challenges of uncertain temporal asynchrony and transmission cost.
- We introduce a self-supervised approach to train the feature flow generator, enabling FFNet with feature prediction ability to mitigate temporal fusion errors across various latencies. This training is independent of cooperative view and labeling, allowing full utilization of infrastructure sequences.
- We evaluate the proposed FFNet on the DAIR-V2X dataset, demonstrating superior performance compared to all cooperative methods while requiring only about 1/100 of the transmission cost of raw data. Furthermore, FFNet is robust across various latencies, requiring only one model.

## 2 Related Work

**Egocentric 3D Object Detection.** Perceiving objects, especially 3D obstacles in the road environment, is a fundamental task in egocentric autonomous driving. Egocentric 3D object detection can be classified into three categories based on sensor types: Camera-based methods, LiDAR-based methods, and multi-sensor-based methods. Camera-based methods, such as FCOS3D (29), directly detect 3D bounding boxes from a single image. BEVformer (21) and $M^2$BEV (31), project 2D images onto a bird's-eye view (BEV) to conduct multi-camera joint 3D detection. LiDAR-based methods, such as VoxelNet (43), SECOND (37), and PointPillars (16), divide the LiDAR point cloud into voxels or pillars and extract features from them. Multi-sensor-based methods (28; 23) utilize both Camera and LiDAR data. In contrast to these methods for single-vehicle view object detection, our proposed method focuses on cooperative detection with point clouds as inputs. It utilizes both infrastructure and vehicle sensor data to overcome the perception limitations of single-vehicle view detection.

**VIC3D Object Detection.** With the development of V2X communication (12), utilizing information from the road environment has attracted much attention. Several works, such as V2VNet (30), DiscoNet (19), StarNet (20) and SyncNet (17), utilize information from other vehicles to expand the perception field. V2X-Sim (18), OPV2V (34) and V2V4Real (32) are datasets for multi-vehicle cooperative perception research. ControllingNet (27) and Coopernaut (6) integrate infrastructure data for end-to-end autonomous driving. Some works like Rope3D (38), BEVHeight (36), and A9-Dataset (5) that focus on utilizing roadside sensor data for 3D object detection. DAIR-V2X (40) is a pioneering work in vehicle-infrastructure cooperative 3D object detection, which introduces the VIC3D object detection task and provides early and late fusion baselines. Then V2X-Seq (42) extends the tasks into cooperative tracking and motion forecasting. Existing approaches such as (14; 2; 15; 8) focus on transmitting feature maps or queries for cooperative detection, without considering the challenges of temporal asynchrony. In this paper, we propose a flow-based feature fusion framework to address the issue of temporal asynchrony and reduce transmission costs in a simple and unified manner. It is important to note that our method is fundamentally different from SyncNet (17), which transmits common features and integrates per-frame features to compensate for latency.

**Feature Flow.** Flow is a concept originating from mathematics, which formalizes the idea of the motion of points over time (7). It has been successfully applied to many computer vision tasks, such as optical flow (3), scene flow (26), and video recognition (45). As a concept extended from optical flow (13), feature flow describes the changing of feature maps over time, and it has been widely used in various video understanding tasks. Zhu et al.(44) propose a flow-guided feature aggregation to improve video detection accuracy. In this paper, we introduce the feature flow for feature prediction to overcome the challenge of temporal asynchrony in VIC3D object detection.

## 3 Method

In this section, we present the proposed FFNet (Feature Flow Net) to solve vehicle-infrastructure cooperative 3D (VIC3D) object detection. We begin by introducing the VIC3D problem in Section 3.1, then explaining the inference process in Section 3.2, and explaining the training methodology of FFNet, including the incorporation of self-supervised learning, in Section 3.3. In the Appendix, we provide a comprehensive comparison of various potential solutions for reference.

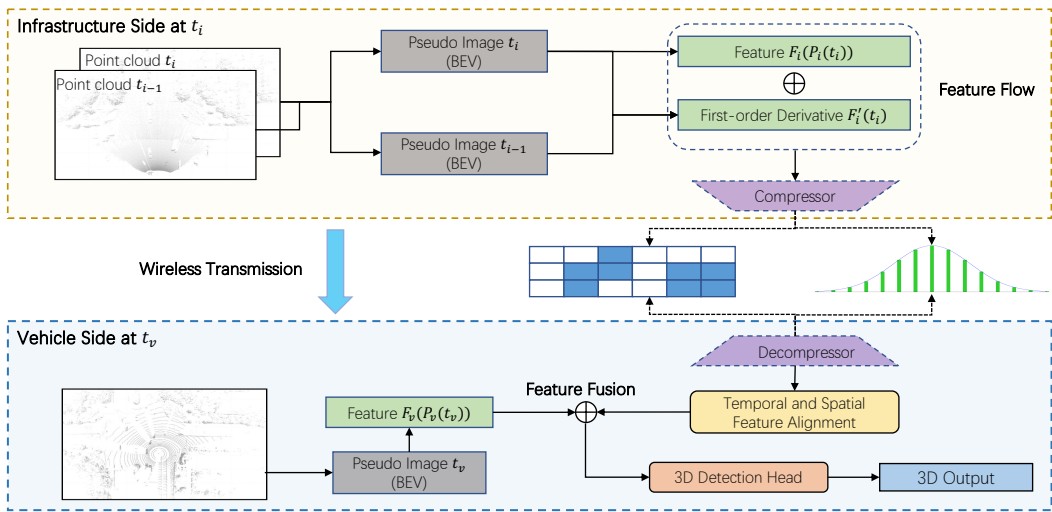

Figure 3: **FFNet Overview.** In the infrastructure system, we represent the feature flow using linear forms by extracting both the feature and the first-order derivative, as shown in Equation 2. To further reduce the transmission cost, we employ attention masks and quantization techniques in addition to a common compressor to compress the feature flow. In the vehicle system, we utilize the feature flow to generate temporally and spatially aligned features. These aligned features are then fused with the vehicle feature to obtain 3D outputs.

## 3.1 VIC3D Object Detection

**Problem Definition.** The VIC3D object detection aims to improve the performance of localizing and recognizing the surrounding objects by utilizing both the infrastructure and vehicle sensor data under limited wireless communication conditions. This paper focuses on point clouds captured from LiDAR as inputs. The input of VIC3D consists of two parts:

- Point cloud $P_v(t_v)$ captured by the ego-vehicle sensor with timestamp $t_v$ as well as its relative pose $M_v(t_v)$, where $P_v(\cdot)$ denotes the capturing function of ego-vehicle LiDAR.
- Point cloud $P_i(t_i)$ captured by the infrastructure sensor with timestamp $t_i$ as well as its relative pose $M_i(t_i)$, where $P_i(\cdot)$ denotes the capturing function of infrastructure LiDAR. Previous frames captured by the infrastructure sensor can also be utilized in cooperative detection.

Note that the timestamp $t_i$ should be earlier than timestamp $t_v$ since receiving the data through long-range communication from infrastructure devices to vehicle devices requires a significant amount of transmission time. Moreover, the latency $(t_v - t_i)$ should be uncertain before receiving the data, as the transmitted data could be obtained by various autonomous driving vehicles in different locations after data broadcasting. The illustration of the uncertain latency is also provided in Figure 2.

**Challenges.** Compared to 3D object detection in single-vehicle autonomous driving scenarios, VIC3D object detection encounters additional challenges related to temporal asynchrony and transmission cost. Directly fusing infrastructure data can lead to significant fusion errors and negatively impact detection performance due to scene changes and the movement of dynamic objects. This asynchronous behavior is show in Figure 2 and evident in the experimental results presented in Section 4.3. Moreover, reducing the amount of transmitted data can effectively decrease the overall latency, as the transmission time is directly influenced by the volume of data being transmitted (9).

**Evaluation Metrics.** We evaluate the 3D object detection performance using mean Average Precision (mAP) with cooperative annotations as the ground truth, as outlined in (10). To focus on the egocentric surroundings, objects outside the designated evaluation area are excluded. For measuring the transmission cost, we adopt the Average Byte ($\mathcal{AB}$) metric, as suggested in (40). The detailed explanations of these two metrics and the computation of $\mathcal{AB}$ are provided in the Appendix.

## 3.2 Feature Flow Net

As depicted in Figure 3, Feature Flow Net (FFNet) consists of three main modules: (1) generating the feature flow, (2) compressing, transmitting, and decompressing the feature flow, and (3) fusing the feature flow with vehicle feature to generate the detection results.

**Feature Flow Generation.** We adopt the feature flow as a prediction function to describe the infrastructure feature changes over time in the future. Given the current point cloud frame $P_i(t_i)$ and the infrastructure feature extractor $F_i(\cdot)$, the feature flow over the future time $t$ after $t_i$ is defined as:

$$\widetilde{F}_i(t) = F_i(P_i(t)), t \geq t_i. \tag{1}$$

Compared with the previous approaches of transmitting per-frame feature $F_i(P_i(t_i))$ produced from per frames (30), which lacks temporal and predictive information, feature flow enables the direct prediction of the aligned feature at the timestamp $t_v$ of the vehicle sensor data.

Two issues need to be addressed in order to apply the feature flow to transmission and cooperative detection: expressing and transmitting the continuous feature flow changes over time, and enabling the feature flow with prediction ability. Considering that the time interval $t_v \rightarrow t_i$ is generally short, we address the expressing issue by using the simplest first-order expansion to represent the continuous feature flow over time, which takes the form of Equation (2),

$$\widetilde{F}_i(t_i + \Delta t) \approx F_i(P_i(t_i)) + \Delta t * \widetilde{F}_i'(t_i), \tag{2}$$

where $\widetilde{F}_i'(t_i)$ denotes the first-order derivative of the feature flow and $\Delta t$ denotes a short time period in the future. Thus, we only need to obtain the feature $F_i(P_i(t_i))$ and the first-order derivative of the feature flow $\widetilde{F}_i'(t_i)$ to approximate the feature flow. When an autonomous driving vehicle receives $F_i(P_i(t_i))$ and $\widetilde{F}_i'(t_i)$ after an uncertain latency, we can generate the infrastructure feature aligned with the vehicle sensor data with minor computation because it only needs linear calculation. To enable the feature flow with prediction ability, we use a network to extract the first-order derivative of the feature flow $\widetilde{F}_i'(t_i)$ from the historical infrastructure frames $I_i(t_i - N + 1), \cdots, I_i(t_i - 1), I_i(t_i)$. Generally, the larger $N$ will generate more accurate estimations. In this paper, we take $N$ as two and use two consecutive infrastructure frames $P_i(t_i - 1)$ and $P_i(t_i)$.

Specifically, we first use the Pillar Feature Net (16) to convert the two consecutive point clouds into two pseudo-images with a bird-eye view (BEV) and with the size of $[384, 288, 288]$. Then, we concatenate the two BEV pseudo-images into the size of $[768, 288, 288]$, and input the concatenated pseudo-images into a 13-layer Backbone and a 3-layer FPN (Feature Pyramid Network), as in SECOND (35), to generate the estimated first-order derivative $\widetilde{F}_i'(t_i)$ with the size of $[364, 288, 288]$. The detailed network configuration is provided in the Appendix.

**Compression, Transmission and Decompression.** In order to eliminate redundant information and reduce the transmission cost, we apply two compressors to the feature $F_i(P_i(t_i))$ and the derivative $\widetilde{F}_i'(t_i)$, compressing them from size $[384, 288, 288]$ to $[12, 36, 36]$ using three Conv-Bn-ReLU blocks in each compressor. We broadcast the compressed feature flow along with the corresponding timestamp and calibration file on the infrastructure side. Upon receiving the compressed feature flow, the vehicle uses two decompressors, each composed of three Deconv-Bn-ReLU blocks, to decompress the compressed feature and compressed first-order derivatives to the original size $[384, 288, 288]$.

We incorporate optional attention masks and quantization techniques to further compress the feature flow. Firstly, we use an attention mask to identify regions of interest and transmit the complete feature along with only the first-order derivative of the feature flow within these regions. Since the infrastructure sensors have fixed positions, the correspondence between elements in the infrastructure feature and real physical space remains constant. The most significant changes in the feature flow over time occur in regions where dynamic instances are moving. To capture these regions, we employ a binary attention mask $M$ that predicts potential dynamic instance locations in the near future. We transmit the feature flow multiplied element-wise by the attention mask, denoted as $M \odot \widetilde{F}_i'(t_i)$, where $\odot$ represents the element-wise product. Secondly, we apply quantization to both the feature and the first-order derivative, reducing them to $b$-bit representations using a linear quantization approach. The quantization is performed according to the following equation:

$$Q(x; \alpha) = \left[\frac{\text{clamp}(x, \alpha)}{s(\alpha)}\right] \cdot s(\alpha), \tag{3}$$

where clamp$(\cdot, \alpha)$ truncates values to the range $[-\alpha, \alpha]$, $[\cdot]$ denotes rounding, and $\alpha$ is the clipping value. We set $\alpha$ as the maximum value of the input tensor, as larger values tend to contain more valuable information (39; 41; 11). We determine $s(\alpha)$ as $\frac{\alpha}{2^{b-1}-1}$. By transmitting $b$-bit numbers instead of the original 32-bit floating-point values and transmitting data only within the regions of interest, we achieve more compression in the data transmission process.

**Vehicle-Infrastructure Feature Fusion.** We use the feature flow to predict the infrastructure feature at timestamp $t_v$, aligned with the vehicle feature, as follows:

$$\widetilde{F}_i(t_v) \approx F_i(P_i(t_i)) + (t_v - t_i) * \widetilde{F}'_i(t_i). \tag{4}$$

This linear prediction operation effectively compensates for uncertain latency and requires minimal computation. The predicted feature $\widetilde{F}_i(t_v)$ is then transformed into the vehicle coordinate system using the corresponding calibration files. The bird's-eye view of the infrastructure and vehicle features are obtained, both at the vehicle coordinate system, while preserving spatial alignment. The feature located outside the vehicle's interest area is discarded for the infrastructure feature, and empty locations are padded with zero elements.

Subsequently, we concatenate the infrastructure and vehicle feature and employ a Conv-Bn-Relu block to fuse the concatenated features. Finally, we input the fused feature into a 3D detection head, utilizing the Single Shot Detector (SSD) (22) setup as the 3D object detection head, to generate 3D outputs for more accurate localization and recognition. The experimental results indicate that the infrastructure feature flow significantly enhances the detection ability.

### 3.3 Training Feature Flow Net

The FFNet training consists of two stages: training a basic fusion framework in an end-to-end way and then using a self-supervised learning to train the feature flow generator.

In the first stage, we train a basic fusion framework in an end-to-end manner without considering latency. This stage aims at enabling FFNet fusing the infrastructure feature with the vehicle feature to enhance detection performance. Specifically, we train FFNet using cooperative data and annotations obtained from both the vehicle and the infrastructure. The localization regression and object classification loss functions used in SECOND (35) are applied in this stage.

In the second stage, we use self-supervised learning to train the feature flow generator by exploiting the temporal correlations in infrastructure sequences, as shown in Figure 4. The idea is to construct the ground truth features by using nearby infrastructure frames that do not require any manual annotations. Specifically, we generate training frame pairs $\mathcal{D} = \{d_{t_i,k} = (P_i(t_i-1), P_i(t_i), P_i(t_i+k))\}$, where $P_i(t_i-1)$ and $P_i(t_i)$ are two consecutive infrastructure point cloud frames, and $P_i(t_i+k)$ is the $(k+1)$-th frame after $P_i(t_i)$.

We construct the loss function to optimize the feature flow generator. The objective is to generate the feature flow to predict $\widetilde{F}_i(t_i+k)$ as close as possible to $F_i(P_i(t_i+k))$. We use the cosine similarity to measure the similarity between the predicted feature and the ground truth feature as

$$similarity = \frac{\widetilde{F}_i(t_i+k) \odot F_i(P_i(t_i+k))}{||\widetilde{F}_i(t_i+k)||_2 * ||F_i(P_i(t_i+k))||_2}, \tag{5}$$

where $\odot$ denotes the inner product, $*$ denotes the scalar multiplication, and $||\cdot||_2$ denotes the L2 norm. We use this similarity as the loss function to train the feature flow generator as

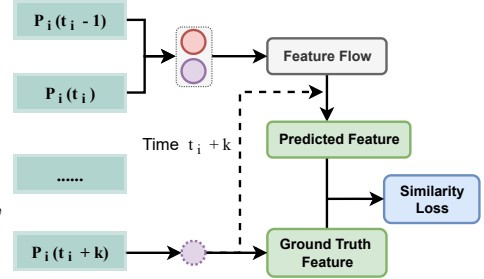

Figure 4: Illustration of training a feature flow generator using self-supervised learning and similarity loss. The upper red circle represents the first-order derivative generator, while the lower purple circles with solid and dashed lines share the same infrastructure feature extractor.

$$\mathcal{L}(\mathcal{D}, \theta) = \sum_{d_{t_i,k} \in \mathcal{D}} (1 - \frac{\widetilde{F}_i(t_i+k) \odot F_i(P_i(t_i+k))}{||\widetilde{F}_i(t_i+k)||_2 * ||F_i(P_i(t_i+k))||_2}), \tag{6}$$

where $\theta$ is the parameter of the feature flow generator, and we only update the parameters in first-order derivative generator $\widetilde{F}'_i(\cdot)$ and frozen other parameters.

# 4    Experiments

In this section, we implement FFNet on the DAIR-V2X dataset (40), comparing it with existing cooperative detection methods on different latencies. Our proposed FFNet outperforms all other methods, including early fusion, V2VNet (30), and DiscoNet (19), at $200ms$ latency, while only requiring about 1/100 of the transmission cost of raw point clouds. We demonstrate how FFNet overcomes the challenge of temporal asynchrony with feature flow prediction. Our results show that temporal asynchrony significantly reduces the performance of the cooperative detection model, but feature flow can effectively compensate for this drop. We evaluate FFNet on different latencies and show that it can robustly solve uncertain latency challenges with just one model. Furthermore, we show that self-supervised training can utilize extra infrastructure sequences. In the Appendix, we compare the performance of feature flow extraction on the infrastructure and ego vehicle sides.

## 4.1    Experiment Settings

**Dataset.**    We used public and real-world DAIR-V2X dataset (40), which comprises over 100 scenes and 18,000 data pairs captured from infrastructure and vehicle sensors (Cameras and LiDARs) at 28 challenging traffic intersections. The dataset includes cooperative 3D annotations with vehicle-infrastructure cooperative view for 9,311 pairs, where each object is labeled with its corresponding category (*Car*, *Bus*, *Truck*, or *Van*). The dataset is divided into *train/val/test* sets in a 5:2:3 ratio, with all models evaluated on the *val* set. Additionally, raw sensor data is only released for the test set.

Note that the timestamps of the data from infrastructure and vehicle sensors in each pair are not precisely synchronized. The time difference of each pair in 9,311 pairs, is within the range of [-30, 30]$ms$. As the actual data cannot be altered after collection, we simulate a latency of $k * 100ms$ by replacing the first $k$ frames of the infrastructure frame with current infrastructure frame for each pair.

**Implementation details.**    We utilized MMDetection3D (1) as our codebase and trained the feature fusion base model on the DAIR-V2X training set for 40 epochs, with a learning rate of 0.001 and weight decay of 0.01. To form $\mathcal{D}$ for training the feature flow generator, we select each pair from training part and randomly set $k$ from the range [1, 2]. More information on $k$ and $\mathcal{D}$ can be found in Sec.3.3. The pretraining of FFNet was done using the trained feature fusion base model. We trained the feature flow generator on $D_u$ for 10 epochs with a learning rate of 0.001 and weight decay of 0.01. All training and evaluation were performed on an NVIDIA GeForce RTX 3090 GPU. The detection performance was measured using KITTI(10) evaluation detection metrics, which include bird-eye view (BEV) mAP and 3D mAP with 0.5 IoU and 0.7 IoU, respectively. Only the *Car* class was taken into account for objects located in the rectangular area [0, -39.12, 100, 39.12]. More implementation details regarding FFNet and the fusion methods are provided in the Appendix.

## 4.2    Comparison to Different Fusion and State of the Art Methods

We compare FFNet with four categories of fusion methods: non-fusion (e.g., PointPillars (16) and AutoAlignV2 (4)), early fusion, late fusion, middle fusion (e.g., DiscoNet (19)), and V2VNet (18).

**Result Analysis.**    Table 1 presents a summary of our experimental results. The table is divided into three parts: the top section displays the evaluation results for non-fusion methods, the middle section shows the results for $200ms$ latency, and the bottom section presents the results for $300ms$ latency. Our proposed FFNet achieves new SOTA on DAIR-V2X. Notably, FFNet-C1 surpasses early fusion while it only requires about 1/100 of the transmission cost. Firstly, our proposed FFNet outperforms the non-fusion method PointPillars by 9.32% mAP@BEV (IoU=0.5) and 7.32% mAP@BEV (IoU=0.5) in $200ms$ and $300ms$ latency, respectively. This result indicates that utilizing infrastructure data can improve 3D detection performance. Secondly, although late fusion requires little transmission cost, the mAP@BEV (IoU=0.5) of late fusion is much lower than that of FFNet, up to 5.10% in $200ms$ latency. Thirdly, compared with early fusion methods, FFNet achieves similar detection performance in $200ms$ latency and outperforms 2.92% mAP in $300ms$ latency, while it only requires no more than 1/10 of the transmission cost. Moreover, FFNet-C1 outperforms early fusion more than 2% mAP in $300ms$ latency while only requiring 1/100 of the transmission cost. Fourthly, our FFNet achieves the best detection performance with the exact transmission cost as the

Table 1: **Comparison to Different Fusion Methods.** FFNet significantly outperforms all other fusion methods.

| Model | FusionType | Latency (ms) | mAP@3D ↑ | | mAP@BEV ↑ | | $\mathcal{AB}$ (Byte) ↓ |
|---|---|---|---|---|---|---|---|
| | | | IoU=0.5 | IoU=0.7 | IoU=0.5 | IoU=0.7 | |
| PointPillars (16) | non-fusion | / | 48.06 | - | 52.24 | - | 0 |
| AutoAlignV2 (4) | non-fusion | / | 50.32 | - | 53.88 | - | 0 |
| Early Fusion | early | 200 | 54.63 | 38.23 | 61.08 | 50.06 | $1.4 \times 10^6$ |
| Late Fusion | late | 200 | 52.43 | 36.54 | 58.10 | 49.25 | $\mathbf{5.1 \times 10^2}$ |
| DiscoNet (19) | middle | 200 | 50.76 | 28.57 | 58.20 | 48.90 | $1.2 \times 10^5$ |
| V2VNet (30) | middle | 200 | 49.67 | 26.96 | 56.02 | 46.32 | $1.2 \times 10^5$ |
| **FFNet (Ours)** | middle | 200 | 55.37 | 31.66 | **63.20** (+9.32) | 54.69 | $1.2 \times 10^5$ |
| **FFNet-C1 (Ours)** | middle | 200 | 55.17 | 31.20 | **62.87** (+8.99) | 54.28 | $\mathbf{1.7 \times 10^4}$ |
| Early Fusion | early | 300 | 51.37 | 37.25 | 58.28 | 49.81 | $1.4 \times 10^6$ |
| Late Fusion | late | 300 | 51.35 | 36.24 | 56.89 | 48.79 | $\mathbf{5.1 \times 10^2}$ |
| DiscoNet (19) | middle | 300 | 49.03 | 27.39 | 55.81 | 47.28 | $1.2 \times 10^5$ |
| V2VNet (30) | middle | 300 | 48.51 | 27.00 | 55.81 | 46.32 | $1.2 \times 10^5$ |
| **FFNet (Ours)** | middle | 300 | 53.46 | 30.42 | **61.20** (+7.32) | 52.44 | $1.2 \times 10^5$ |
| **FFNet-C1 (Ours)** | middle | 300 | 54.10 | 29..87 | **60.76** (+6.88) | 53.28 | $\mathbf{1.7 \times 10^4}$ |

Table 2: **Comparison between with and without Feature Prediction.** Compared with no prediction models, FFNet with feature prediction has a significantly lower performance drop when there is communication latency.

| Model | Latency (ms) | mAP@3D ↑ | | mAP@BEV ↑ | | AB (Byte) ↓ |
|---|---|---|---|---|---|---|
| | | IoU=0.5 | IoU=0.7 | IoU=0.5 | IoU=0.7 | |
| FFNet | 0 | 55.81 | 30.23 | 63.54 | 54.16 | $1.2 \times 10^5$ |
| FFNet (without prediction) | 0 | 55.81 | 30.23 | 63.54 | 54.16 | $6.2 \times 10^4$ |
| FFNet-V2 (without prediction) | 0 | 55.78 | 30.22 | 64.23 | 55.00 | $1.2 \times 10^5$ |
| FFNet | 200 | 55.37 | 31.66 | 63.20 (-0.34) | 54.69 | $1.2 \times 10^5$ |
| FFNet (without prediction) | 200 | 50.27 | 27.57 | 57.93 (-5.61) | 48.16 | $6.2 \times 10^4$ |
| FFNet-V2 (without prediction) | 200 | 49.90 | 27.33 | 58.00 (-6.23) | 48.22 | $1.2 \times 10^5$ |

middle fusion methods. For example, FFNet surpasses DiscoNet by 5.0% mAP@BEV (IoU=0.5) and 5.39% mAP@BEV (IoU=0.5) in $200ms$ and $300ms$ latency, respectively.

## 4.3  Ablation Study

We conducted a series of experiments to demonstrate the effectiveness of the feature flow module in overcoming the temporal asynchrony challenge and to show that FFNet performs robustly under various latencies. Additionally, we studied how self-supervised learning can fully exploit infrastructure sequences that are independent of cooperative view and labeling.

**Feature prediction can well solve temporal asynchrony.**   We conducted a series of experiments to evaluate the effectiveness of the feature flow module in overcoming the temporal asynchrony challenge. We evaluated FFNet under two different latency conditions: $0ms$ and $200ms$, where $0ms$ indicates temporal asynchrony between infrastructure data and vehicle data within $[-30, 30]ms$. To investigate the impact of temporal asynchrony on FFNet's performance, we also removed the prediction module from FFNet and directly fused the infrastructure feature. We refer to this version as FFNet (without prediction), abbreviated as FFNet-O, and evaluated it under both $0ms$ and $200ms$ latency. Since FFNet-O does not require the transmission of the first-order derivative of the feature flow, it only requires half the transmission cost of FFNet. To ensure a fair comparison, we trained another version of FFNet called FFNet-V2, which compressed the feature flow from (384, 288, 288) to (384/16, 288/8, 288/8). FFNet-V2-O has the same transmission cost as FFNet, and we evaluated it under both $0ms$ and $200ms$ latency as well.

The evaluation results, presented in Table 2, demonstrate that FFNet-O and FFNet-V2-O exhibit a significant performance drop under $200ms$ latency. For example, FFNet-O experiences a 5.61% mAP@BEV (IoU=0.5) drop in $200ms$ latency compared to $0ms$ latency. Although FFNet-V2-O performs slightly better than FFNet and FFNet-O in $0ms$ latency, FFNet significantly outperforms FFNet-V2-O in $200ms$ latency. These results show that temporal asynchrony can significantly impact performance when we directly fuse the infrastructure feature, and that our feature prediction module can effectively compensate for the performance drop caused by temporal asynchrony.

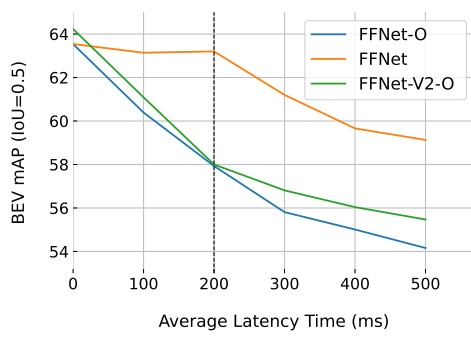

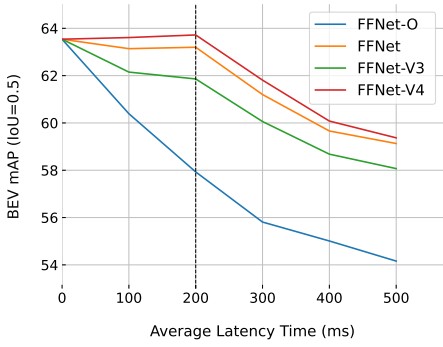

Figure 5: Ablation study of FFNet robustness.      Figure 6: Ablation study of FFNet training.

**FFNet is robust to uncertain latency.** We conducted additional experiments to assess the performance of FFNet, FFNet-O, and FFNet-V2-O under varying latency cases, ranging from $100ms$ to $500ms$. The experiment results are presented in Figure 5. As depicted in the figure, both FFNet-O and FFNet-V2-O exhibit continuous performance degradation as the latency increases from $100ms$ to $500ms$. Specifically, in $500ms$ latency, FFNet-O and FFNet-V2-O show a significant 9.38% mAP@BEV (IoU=0.5) drop and 9.76% mAP@BEV (IoU=0.5) drop, respectively. Conversely, FFNet demonstrates minimal performance degradation within $200ms$ latency and only experiences a 4.39% mAP@BEV (IoU=0.5) drop. These results suggest that FFNet is resilient to varying latencies and can effectively handle uncertain latency in VIC3D problem. The ability of our feature flow to make predictions at an arbitrary future time before transmission is crucial since it could be received by different vehicles with different latencies.

**FFNet training can fully utilize infrastructure sequences.** We additionally trained the feature flow generator using the extra test portion of the DAIR-V2X dataset. For each frame in the test part, we randomly set $k$ from the range [1, 2] to form $\mathcal{D}_{test}$. We first pretrained FFNet with the trained feature fusion base model and trained the feature flow generator solely with $\mathcal{D}_{test}$ without $\mathcal{D}$. We refer to this trained FFNet as FFNet-V3. Subsequently, we pretrained FFNet with the trained feature fusion base model and trained the feature flow generator using $\mathcal{D}_{test} \cup \mathcal{D}$. We denote this trained FFNet as FFNet-V4. We evaluated FFNet-V3 and FFNet-V4 under latency from $100ms$ to $500ms$. In Figure 5, we present the mAP@BEV (IoU=0.5) results. FFNet-V3 demonstrates significantly better performance than FFNet-O, indicating that the training of the feature flow generator can be independent of cooperative-view data. FFNet-V4 performs slightly better than FFNet, suggesting that incorporating more infrastructure sequences enhances the feature flow prediction ability.

## 5 Conclusion

This paper introduces FFNet, an innovative intermediate-level cooperative framework designed for VIC3D object detection. FFNet effectively addresses challenges related to temporal asynchrony and transmission cost by utilizing compressed feature flow for cooperative detection. Through extensive experiments conducted on the DAIR-V2X dataset, FFNet demonstrates superior performance compared to existing state-of-the-art methods. Furthermore, FFNet can be extended to various modalities, including image and multi-modality data, making it a versatile solution. Moreover, FFNet holds promise in the domain of multi-vehicle cooperative perception and leverages the utilization of additional frames to enhance feature prediction capabilities. The proposed FFNet framework, incorporating feature prediction and self-supervised learning, presents a promising avenue for VIC3D object detection and holds potential for addressing diverse cooperative perception tasks in the future.

## Acknowledgements

This paper is partially supported by the National Key R&D Program of China No.2022ZD0161000 and the General Research Fund of Hong Kong No.17200622. This work was also supported by Baidu Inc. through the Apollo-AIR Joint Research Center.

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
