# Flow-based Feature Fusion for Vehicle-Infrastructure Cooperative 3D Object Detection – Appendix

**Haibao Yu**[1,2], **Yingjuan Tang**[2,3], **Enze Xie**[1], **Jilei Mao**[2], **Ping Luo**[1,4], **Zaiqing Nie**[2] *
[1]The University of Hong Kong
[2]Institute for AI Industry Research (AIR), Tsinghua University
[3]Beijing Institute of Technology [4]Shanghai AI Laboratory

## A    Detailed Evaluation Metrics for VIC3D Object Detection

In this section, we detail the evaluation metrics used to assess the performance of the VIC3D object detection algorithm, namely mean Average Precision (mAP) and Average Byte ($\mathcal{AB}$).

**Mean Average Precision (mAP).**    We evaluate the detection performance using the mean Average Precision (mAP) metric, which is commonly used in previous works such as (10; 1). The AP is calculated based on the 11-points interpolated precision-recall curve and is defined as follows:

$$AP = \frac{1}{11} \sum_{r \in 0.0,...,1.0} AP_r \quad = \frac{1}{11} \sum_{r \in 0.0,...,1.0} P_{interp}(r), \tag{1}$$

where $P_{interp}(r) = \max_{\tilde{r} \geq r} p(\tilde{r})$, and a prediction is considered positive if the Intersection over Union (IoU) is greater than or equal to 0.5 or 0.7, respectively. We calculate the AP for each class and then average them to obtain the mAP.

For VIC3D object detection, we focus on the obstacles around the ego vehicle. Therefore, we only consider objects within the regions of interest around the ego vehicle. To evaluate the detection performance, we transform all predicted 3D boxes and ground truth 3D boxes into the ego-vehicle coordinate system. To define the regions of interest, we remove objects outside the egocentric surroundings, which are defined as a rectangular area with coordinates [0, -39.12, 100, 39.12].

There are two metrics used for evaluation: BEV@mAP and 3D@mAP. BEV@mAP evaluates the 3D boxes in the bird's-eye view and ignores the $z$-dimension, while 3D@mAP considers all three dimensions $(x, y, z)$ and is more strict than BEV@mAP.

**Average Byte ($\mathcal{AB}$).**    We use $\mathcal{AB}$ as a metric to evaluate the transmission cost. In our implementation, we ignore the transmission cost of calibration files and timestamps. The average transmission cost is computed as $\mathcal{AB}$, and we explain how to calculate the transmission cost for each of the three transmission forms:

- For early fusion, we calculate the transmission cost of transmitting raw data. Each point in the point clouds is represented as $(x, y, z, \text{intensity})$ in 32-bit float format. Therefore, each point requires four 32-bit floats, equivalent to 16 Bytes. If there are 100,000 points in the point clouds per transmission, the $\mathcal{AB}$ of the transmission cost is $1.6 \times 10^6$ Bytes.

- For late fusion, we calculate the transmission cost of transmitting detection outputs. Each 3D detection output is represented as $(x, y, z, w, l, h, \theta, \text{confidence})$ in 32-bit float format. Thus, each detection output requires eight 32-bit floats, equivalent to 32 Bytes. If we transmit ten detection outputs per transmission, the $\mathcal{AB}$ of the transmission cost is $3.2 \times 10^2$ Bytes.

---

*Corresponding author. Work done while at AIR.

37th Conference on Neural Information Processing Systems (NeurIPS 2023).

- For middle fusion, we calculate the transmission cost of transmitting feature, which is represented as a tensor. If the size of the feature is $(24, 36, 36)$ and each element is encoded as a 32-bit float, the transmission cost is $24 \times 36 \times 36 \times 4$ Bytes, amounting to $1.2 \times 10^5$ Bytes.

- For middle fusion, we also calculate the transmission cost of transmitting feature flow. Both feature and the first-order derivative of feature flow are represented as tensors. If the size of the feature and the first-order derivative is $(12, 36, 36)$ respectively, with each element encoded as a 32-bit float, the transmission cost is $12 \times 36 \times 36 \times 2 \times 4$ Bytes, equivalent to $1.2 \times 10^5$ Bytes. If we quantize the intermediate data into $b$-bit and transmit the quantized feature and first-order derivative, the transmission cost becomes $(1.2 \times 10^5) \times b/32$ Bytes. Further details on the calculation of the transmission cost with the attention mask to compress the feature flow are provided in Section C.

## B    Architecture of FFNet

FFNet is composed of following six main parts.

- The feature flow generation module: The infrastructure PFNet (Pillar Feature Net) shares the same architecture as PointPillars (3). The x, y, and z ranges of the input point cloud are [(0, 92.16), (-46.08, 46.08), (-3, 1)] meters, respectively. The voxel size of x, y, and z are [0.16, 0.16, 4] meters, respectively. The output shape of the pseudo-images is (64, 576, 576). The feature extractor $F_i(\cdot)$ and the estimated first-order derivative generator $\widetilde{F}_i^{'}(\cdot)$ both use the same Backbone and FPN as SECOND (8), with output shapes of [384, 288, 288].

- The compressor and decompressor: The compressor utilizes four convolutional blocks with strides (2, 1, 2, 2) to compress the features from (384, 288, 288) to (384/32, 288/8, 288/8). The decompressor employs three deconvolutional blocks with strides (2, 2, 2) to restore the features back to their original size.

- Affine transformation module: The affine transform is implemented with the $affine\_grid$ function supported in Pytorch. Rotation around the x-y plane is ignored.

- Feature fusion module: The fusion module is a $3 \times 3$ convolutional block with a stride 1 to compress the concatenated feature from (768, 288, 288) to (384, 288, 288).

- Vehicle feature extractor: This extractor follows the same configuration as the infrastructure PFNet and feature extractor.

- 3D object detection head: A Single Shot Detector (SSD) (6) is used to generate the 3D outputs. The anchor has a width, length, and height of (1.6, 3.9, 1.56) meters, with a z-center of -1.78 meters. The positive and negative thresholds of matching are 0.6 and 0.45, respectively.

## C    Attention Mask for Compression

**Implementation Details.**    The attention mask $M$ has the same height and width as the compressed first-order derivative of the feature flow $\widetilde{F}_i^{'}(t_i)$, with a single channel and a size of (36, 36). Each element of the attention mask is obtained by element-wise multiplication between $M$ and $\widetilde{F}i^{'}(t_i)$ across all channels, given by:

$$(M \odot \widetilde{F}_i^{'}(t_i))_{j,k,l} = M_{k,l} * \widetilde{F}_i^{'}(t_i)_{j,k,l}. \tag{2}$$

To generate the attention mask, we compute the feature difference between consecutive pseudo-images. Then, we divide the pseudo-image space into 32x32 patches. For each patch, if the feature difference exceeds a certain threshold, we set the corresponding patch element in the mask to 1; otherwise, it is set to 0. Here we set the threshold as 0.0. Below is a Python implementation code for the process of determining the attention mask:

```python
def AttentionMask(image_1, image_2, img_shape, mask_shape, thre):
    % default parameters
    % img_shape=(576, 576)
    % mask_shape=(36, 36)
    % thre=0.0
    mask = torch.zeros(mask_shape[0], mask_shape[1]))
```

```
    feat_diff = torch.sum(torch.abs(image_1 - image_2), dim=1)
    stride = int(img_shape[0] / mask_shape[0])
    for k in range(mask_shape[0]):
        for l in range(mask_shape[1]):
            patch = feat_diff[k*stride:(k+1)*stride, l*stride:(l+1)*
                                                    stride]
            if patch.sum() > thre:
                mask[k, l] = 1

    return mask
```

**Transmission and Transmission Cost.**   To transmit the first-order derivative with the attention mask, we transmit both the binary mask $M$ and the non-zero elements of $(M \odot \widetilde{F}'_i(t_i))$. The binary mask $M$ is represented using 1 bit per element. The total transmission cost of the binary attention mask is calculated as $(36 \times 36)/8$ Bytes. The non-zero elements of $(M \odot \widetilde{F}'_i(t_i))$ account for the proportion of non-zero elements in the attention mask, denoted as $P$, multiplied by the original transmission cost. Specifically, the transmission cost of the non-zero elements is given by $P \times 12 \times 36 \times 36 \times 4$ Bytes. Here the average proportion of non-zero elements $P$ is about 60%.

## D   Quantization for Compression

**Implementation Details.**   We employ linear quantization to represent the feature flow using $b$-bit. The quantization process follows the equation:

$$Q(x; \alpha) = [\frac{\mathrm{clamp}(x, \alpha)}{s(\alpha)}] \cdot s(\alpha), \tag{3}$$

where $\mathrm{clamp}(\cdot, \alpha)$ truncates values to the range $[-\alpha, \alpha]$, $[\cdot]$ denotes rounding, and $\alpha$ is the clipping value. We set $\alpha$ as the maximum value of the input tensor, as larger values tend to contain more valuable information (9; 11; 2). We determine $s(\alpha)$ as $\frac{\alpha}{2^{b-1}-1}$. Below is a Python implementation code for the quantization process:

```
def QuantFunc(input, b_n=6):
    alpha = torch.abs(input).max()
    s_alpha = alpha / (2 ** (b_n - 1) - 1)

    input = input.clamp(min=-alpha,max=alpha)
    input = torch.round(input/s_alpha)
    input = input * s_alpha

    return input
```

**Transmission.**   During the transmission process, we transmit the rounded number $[\frac{\mathrm{clamp}(x, \alpha)}{s(\alpha)}]$, which effectively represents the quantized feature flow using $b$-bits. Additionally, we transmit the scaling factor $s(\alpha)$ required for decoding the quantized values. In our study, **we set the value of $b$ to 6 bits for quantization.**

**Experiment Results.**   We present the experimental results of FFNet with 6-bit quantization in Table 1. It can be observed that the performance of FFNet with 6-bit quantization exhibits only a minimal decrease compared to the original FFNet without quantization.

## E   Implementation Details of V2VNet and DiscoNet for VIC3D

**V2VNet for VIC3D.**   V2VNet (7) is a pioneering work in multi-vehicle cooperative perception, introducing the concept of transmitting intermediate-level data for cooperative perception without relying on sequential frames to extract temporal correlations. In this paper, we adopt this approach as a baseline for solving the VIC3D problem, as depicted in Figure 1. We made two modifications to

Table 1: **Experimental Results: Quantization Impact. We employ 6-bit quantization.**

| Latency | Quantization | mAP@3D ↑ | | mAP@BEV ↑ | | AB (Byte) ↓ |
|---|---|---|---|---|---|---|
| | | IoU=0.5 | IoU=0.7 | IoU=0.5 | IoU=0.7 | |
| 200 | N | 55.37 | 31.20 | 63.20 | 54.69 | $1.2 \times 10^5$ |
| 200 | Y | 55.39 | 31.69 | 63.26 (+0.06) | 54.63 | $2.2 \times 10^4$ |
| 300 | N | 53.46 | 30.42 | 61.20 | 52.44 | $1.2 \times 10^5$ |
| 300 | Y | 53.37 | 30.43 | 61.28 (+0.08) | 52.40 | $2.2 \times 10^4$ |

the V2VNet architecture: (1) we removed the multi-vehicle selection and kept only one vehicle in the infrastructure setting, and (2) we compressed the features from (384, 288, 288) to (384, 288/8, 288/8) to ensure a comparable transmission cost to FFNet. The remaining modules maintain the same configurations as their corresponding counterparts in FFNet. We trained the V2VNet model on the training subset of the DAIR-V2X dataset for 40 epochs, employing a learning rate of 0.001 and a weight decay of 0.01. The remaining training configurations align with those used for training FFNet.

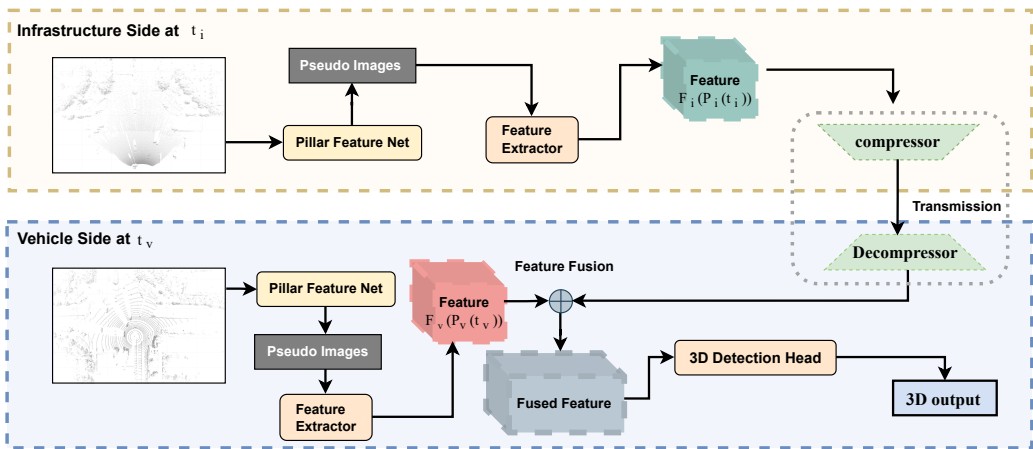

Figure 1: **Implementation of V2VNet for Solving VIC3D Problem.** The V2VNet directly transmit the feature generated from the single point cloud, and then fuse it with the vehicle feature. This feature fusion could cause serious fusion errors by the uncertain temporal asynchrony.

**DiscoNet for VIC3D.**  DiscoNet (5) was originally designed for cooperative perception among multiple vehicles. It utilizes a teacher-student paradigm, where cooperative perception with raw data serves as the teacher network to guide cooperative perception with intermediate data, which acts as the student network. To adapt DiscoNet as a baseline for the VIC3D task, we employ an early-fusion network as the teacher network and V2VNet as the student network. Both the teacher and student models are trained on the training subset of DAIR-V2X for 40 epochs, with a learning rate of 0.001 and weight decay of 0.01. Furthermore, we fine-tune the student network using soft labels generated by the early-fusion network for an additional 10 epochs, with a learning rate of 0.0001 and weight decay of 0.01.

## F    Comparison of Feature Flow Extraction on Different Sides

This section discusses the effect of extracting the feature flow on different sides (infrastructure side *vs.* vehicle side).

**Experiment Setting.**    To compare the effect of feature flow extraction on infrastructure side and vehicle side, we train a modified FFNet called FFNet-V. The FFNet-V inputs the features $F_i(P_i(t_i - 1))$ and $F_i(P_i(t_i - 1))$ produced from consecutive infrastructure frames to generate the feature flow on vehicle devices. We first concatenate the two received features and feed them into a first-order derivative generator to generate the estimated first-order derivative of the feature flow $\widetilde{F_i'}(t_i)$. Then

we predict the future feature as following Equation

$$\widetilde{F}_i(t_i + \Delta t) \approx F_i(P_i(t_i)) + (t_v - t_i) * \widetilde{F}_i^{'}(t_i). \qquad (4)$$

In addition, FFNet-V shares the same architecture modules and training configuration as FFNet. The FFNet-V implementation framework is shown in Figure 2. To ensure a fair comparison with FFNet, we also train another FFNet-V by compressing the feature from (384, 288, 288) to (384/16, 288/8, 288/8), which has the same transmission cost as FFNet. This version of FFNet-V is referred to as FFNet-V (Same-TC). We evaluate both FFNet-V and FFNet-V (Same-TC) under different latencies (100$ms$, 300$ms$, and 500$ms$).

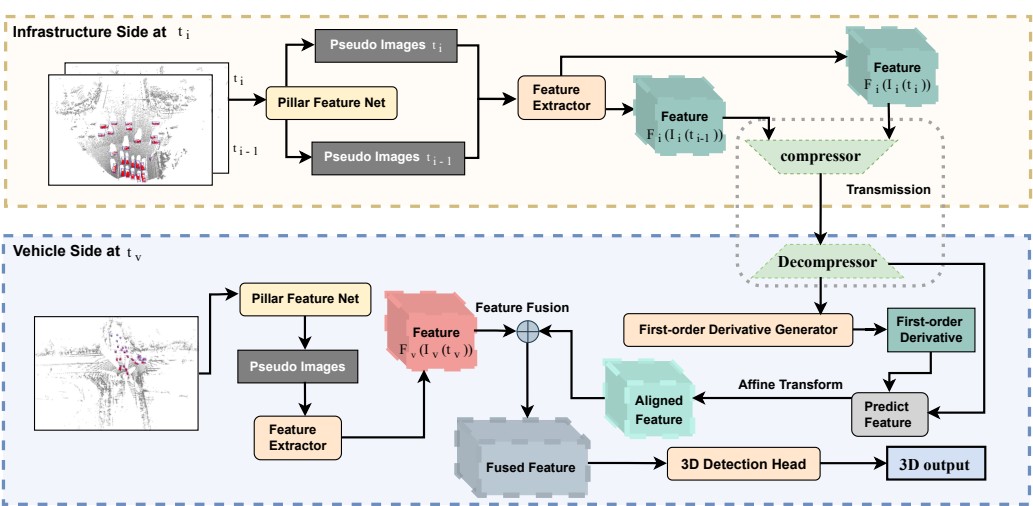

Figure 2: FFNet-V Overview. FFNet-V generates the feature flow on vehicle devices.

**Result Analysis.** Table 2 shows that FFNet-V, FFNet-V (Same-TC), and FFNet outperform FFNet-O under different latencies. This indicates that all these methods can reduce the detection performance drop caused by temporal asynchrony. However, FFNet can compensate for more performance drop and achieve better performance than FFNet-V and FFNet-V (Same-TC). Specifically, FFNet outperforms FFNet-V (Same-TC) by more than 3% mAP@BEV (IoU=0.5) in 300$ms$ latency, while having the same transmission cost. The results demonstrate that extracting feature flow from raw sequential frames on infrastructure can improve the VIC3D detection performance more effectively than extracting feature flow from intermediate sequential feature frames on vehicle. Moreover, FFNet requires much fewer ego-vehicle computing resources, and the computing cost complexity (CCC) is only O(N), since the feature flow has already been produced on infrastructure devices and does not need to be generated on vehicle devices again. In contrast, FFNet-V consumes computation resources up to O(N) to extract flow from past features. Therefore, FFNet is more computation-friendly for resource-limited vehicle devices. Additionally, extracting feature flow on the vehicle requires much more storage because the feature flow extraction depends on the past frames that the vehicle received. Furthermore, FFNet-V relies heavily on past consecutive frames, so dropped frames can significantly affect the execution and performance. Therefore, FFNet is more storage-friendly to the ego vehicle and more robust to frame dropping.

## G   Relationship to Other Existing Possible Solutions

Compared to other solutions, FFNet offers a more practical paradigm for implementing vehicle-infrastructure cooperative 3D object detection, providing the following advantages:

- Performance-Bandwidth Balance: FFNet achieves a superior balance between performance and bandwidth compared to early fusion and late fusion methods. Unlike early fusion, FFNet transmits compressed intermediate data, reducing transmission costs. Additionally, FFNet transmits valuable information for egocentric object detection, surpassing the capabilities of late fusion methods.

Table 2: **Extracting Feature Flow on Infrastructure side *vs.* on Vehicle Side.** FFNet-O denotes the FFNet model without feature prediction. FFNet-V denotes the model that extracts the feature flow on vehicle. FFNet-V (Same-TC) denotes the FFNet-V which has the same transmission cost as FFNet. "AB" denotes the average byte used to measure the transmission cost. "SCC" indicates the storage cost complexity for the vehicle devices to store the past frames, "CCC" indicates the computing cost complexity for vehicle to extract the feature flow, "N" indicates the number of historical structures to be used. "/" indicates the FFNet-O does not need infrastructure transmission and extra computation and storage. The SCC of FFNet is O(1) because it does not need extra historical frames on vehicle devices. At the same time, the SCC of extracting feature flow on vehicle is O(N) because extracting feature flow on vehicle needs past frames received from infrastructure. Moreover, FFNet achieves better detection performs, and this advantage becomes more pronounced (+3% mAP) when latency increases to $300ms$.

| Model | Latency (ms) | mAP@3D ↑ | | mAP@BEV ↑ | | AB(Byte) ↓ | SCC ↓ | CCC ↓ |
|---|---|---|---|---|---|---|---|---|
| | | IoU=0.5 | IoU=0.7 | IoU=0.5 | IoU=0.7 | | | |
| FFNet-O | 100 | 52.18 | 27.99 | 60.39 | 49.14 | / | / | / |
| FFNet-V | 100 | 53.21 | 28.43 | 61.50 | 50.50 | $6.2\times10^4$ | O(N) | O(N) |
| FFNet-V (Same-TC) | 100 | 53.17 | 28.45 | 62.44 | 51.68 | $1.2\times10^5$ | O(N) | O(N) |
| **FFNet (Ours)** | 100 | 55.48 | 31.50 | **63.14** (+0.7) | 54.28 | $1.2\times10^5$ | O(1) | O(1) |
| FFNet-O | 300 | 49.03 | 27.39 | 55.81 | 47.28 | / | / | / |
| FFNet-V | 300 | 50.81 | 28.45 | 57.75 | 49.62 | $6.2\times10^4$ | O(N) | O(N) |
| FFNet-V (Same-TC) | 300 | 50.5 | 28.25 | 58.02 | 50.03 | $1.2\times10^5$ | O(N) | O(N) |
| **FFNet (Ours)** | 300 | 53.46 | 30.42 | **61.20** (+3.18) | 52.44 | $1.2\times10^5$ | O(1) | O(1) |
| FFNet-O | 500 | 47.49 | 27.01 | 54.16 | 45.99 | / | / | / |
| FFNet-V | 500 | 49.93 | 28.63 | 56.42 | 48.87 | $6.2\times10^4$ | O(N) | O(N) |
| FFNet-V (Same-TC) | 500 | 49.98 | 27.7 | 56.99 | 49.55 | $1.2\times10^5$ | O(N) | O(N) |
| **FFNet (Ours)** | 500 | 52.08 | 30.11 | **59.13** (+2.14) | 51.70 | $1.2\times10^5$ | O(1) | O(1) |

- Overcoming the Temporal Asynchrony Challenge: FFNet addresses the challenge of temporal asynchrony between vehicle and infrastructure sensors. Unlike V2VNet (7) and DiscoNet (5), which only transmit features without considering temporal asynchrony, FFNet transmits the feature flow along with feature prediction capabilities. This feature flow generates future features aligned with vehicle features, mitigating fusion errors caused by temporal asynchrony. Notably, the independent module of the first-order derivative in the feature flow can be applied to newer feature fusion methods, achieving lower transmission costs.

- Computing-Friendly for Vehicles with Limited Resources: FFNet generates the feature flow on the infrastructure side and can directly predict future features, compensating for uncertain latency through linear computation in ego vehicles. Another solution proposed in (4) addresses temporal asynchrony by generating future features with received historical features on vehicle devices. However, this solution demands significant computing resources to process historical frames and extract temporal correlations for future feature prediction. Extracting temporal information from compressed features poses challenges, as compressed features lack valuable information present in raw sequential point clouds.

- Annotation Cost Savings: FFNet training significantly reduces annotation costs. A self-supervised learning method is employed to train the feature flow generator and extract temporal feature flow from sequential point clouds. This training method does not rely on labeled data and opens up possibilities for utilizing vast amounts of unlabeled infrastructure-side sequential data in the future.

# H   Visualization Results

**Infrastructure sensor data can broaden the perception field.**   We provide a visualization example in Figure 3 to show that infrastructure sensor data can broaden the perception ability of autonomous driving car.

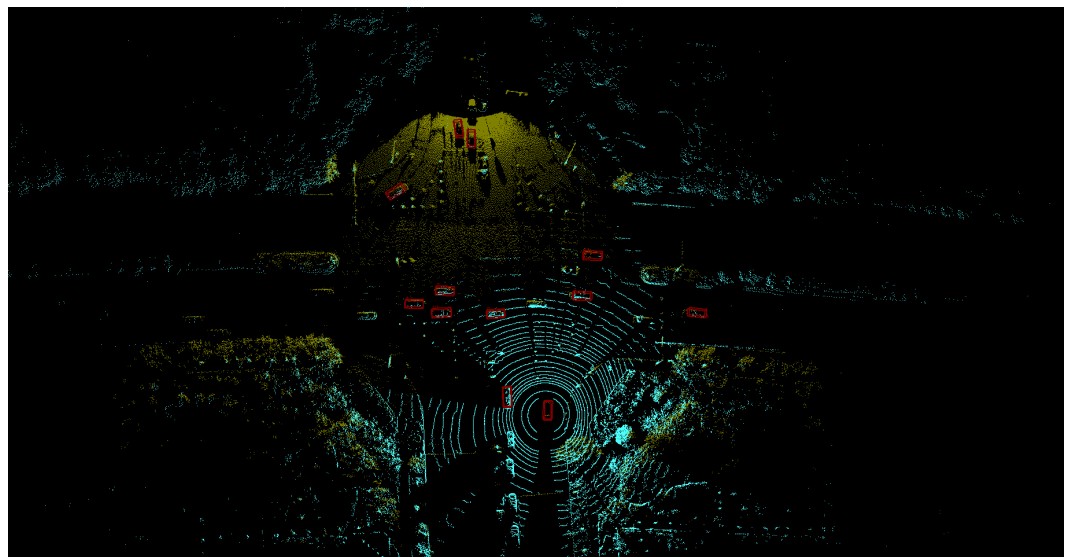

Figure 3: The Effect of Infrastructure Sensor Data. The brown point clouds represent data captured from the infrastructure sensors, the blue point clouds represent data captured from the vehicle sensors, and the red boxes indicate the prediction outputs obtained using FFNet.

**Effect of Feature Flow Prediction.**   We generate detection outputs using feature flow prediction and without feature flow prediction, respectively. To demonstrate the effect of feature flow prediction, we provide visualization examples in Figure 4.

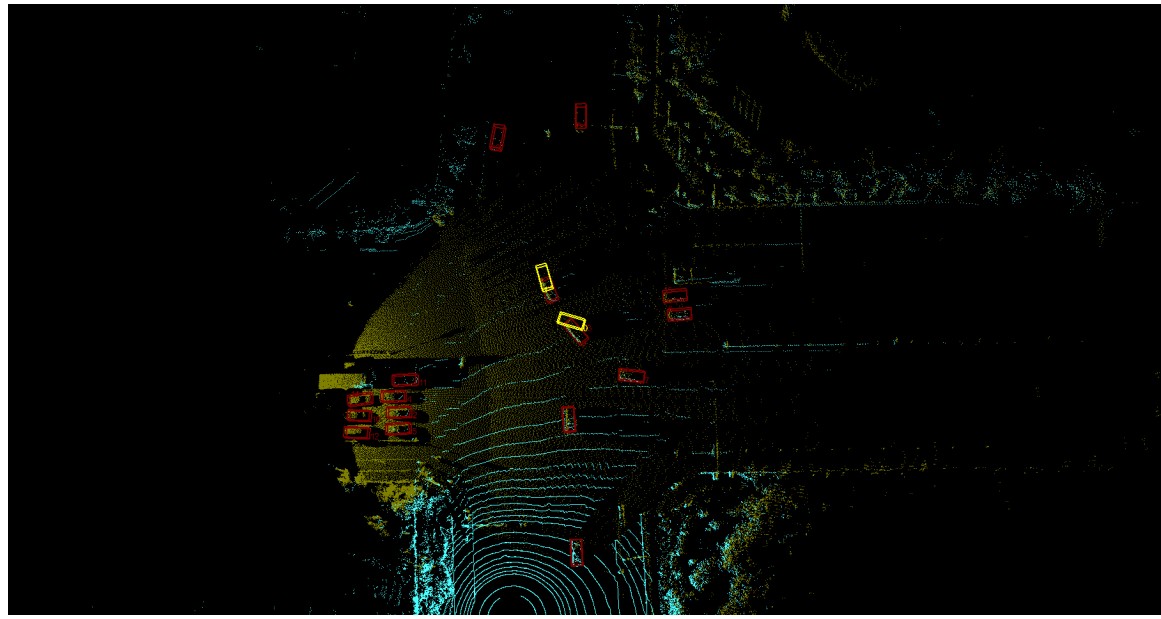

(a) Detection Result of FFNet without Feature Prediction.

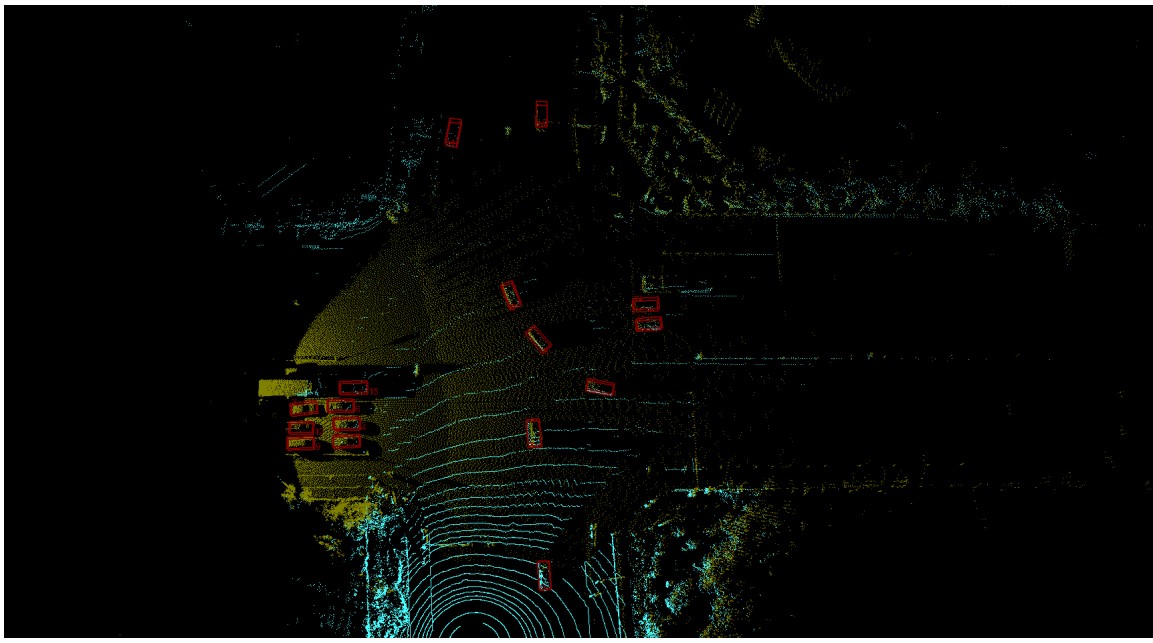

(b) Detection Result of FFNet with Feature Prediction.

Figure 4: **The Effect of Feature Flow Prediction.** The brown point clouds represent data captured from the infrastructure sensors, and the blue point clouds represent data captured from the vehicle sensors. The yellow boxes highlight the additional detection outputs resulting from the non-aligned feature fusion caused by temporal asynchrony.