# OpenReview forum: "Flow-Based Feature Fusion for Vehicle-Infrastructure Cooperative 3D Object Detection"
_NeurIPS.cc/2023/Conference — NeurIPS 2023 poster_

### Official Review · Reviewer_Wmh8 · 2023-07-05

**Soundness:** 2 fair
**Presentation:** 3 good
**Contribution:** 2 fair
**Rating:** 5
**Confidence:** 3

**Summary:**

The authors explore and analyze the existing vehicle-infrastructure cooperative 3D object detection framework, and propose to adopt flow-based rather than still images to extract temporal features from multiple LiDAR input frames. The experimental results on DAIR-V2X dataset is better compared to the existing methods in terms of car objects.

**Strengths:**

1. The task of vehicle-infrastructure cooperative 3D object detection is very important in the 3D community and has real-world practical applications. Interestingly, the authors propose to extract temporal features for different input frames. Also, they propose a self-supervised method to extract the flow-based features.

**Weaknesses:**

1. The evaluation of one single class on a single dataset is not strong enough to prove the effectiveness of the proposed method. It would be convincing if the authors could conduct the experiments on other labeled classes, i.e., bus, truck, or van. Also, experiments on another existing dataset would be better as well.

2. Memory footprint comparison. The authors didn't make a comparison of their work in terms of memory with the existing vehicle-infrastructure cooperative 3D detection methods. As far as I am concerned, the memory consumption/additional efforts are needed since the compression and transmission are included.

**Questions:**

Please refer to the questions that I describe in the Weakness part. I would also consider the rebuttal and other reviews.

---

> ### Author Rebuttal · Authors · 2023-08-09
>
> Dear Reviewer Wmh8,
>
> Thank you for providing valuable feedback on our work. We will address each of the limitations you have pointed out in your comments.
>
>
> **W1. We have conducted experiments on more datasets like OPV2V[1],** which also focuses on cooperative 3D detection. Our forthcoming version will also include experiments conducted on the V2X-Sim[2] dataset.  Notably, the comment at the top offers a concise summary of our experiment results. These results, spanning the DAIR-V2X[3] and OPV2V[1] datasets, show the effectiveness of our proposed method.
>
> **W2. We have augmented our analysis to include memory comparisons with existing cooperative 3D detection methods.** To ensure a focused evaluation, we specifically compare our FFNet with  DiscoNet[4], as the inference network structure of DiscoNet aligns closely with that of V2VNet[5]. Detailed implementation specifics for both methods are meticulously outlined in the supplementary materials (refer to Section E  "Implementation Details of V2VNet and DiscoNet for VIC3D" in  "S1-Appendix.pdf"). This comparison is summarized in Table 5 below. Notably, memory calculations were executed using an NVIDIA A100 GPU. In contrast to DiscoNet[5], **FFNet exhibits a slightly elevated memory footprint due to the processing of multiple frames on the infrastructure side, while the vehicle side memory remains consistent.** However, this increased memory usage remains well within acceptable limits for infrastructure computing servers.
>
> |               | Infrastructure Side |            |            |  | Vehicle Side |            |             |
> |---------------|---------------------|------------|-----------| --|--------------|------------|-------------|
> |               | Feature Flow     Generation      | Compression | Total       |  |     |
> | DiscoNet[5]      | /                   | 122.1M     | 243.9M     |  | 144.3M       |            |             |
> | FFNet         | 486.0M              | 122.1M     | 610.4M     |  | 144.3M       |            |             |
> | FFNet-C1      | 486.0M              | 286.1M     | 610.4M    |   | 144.3M       |            |             |
> | | | | | | | | |
>
> Table 5. Comparison of Memory Footprints. FFNet-C1 refers to FFNet with additional proposed compression modules.
>
> Furthermore, autonomous vehicles, with limited resources, are more sensitive to the memory footprint and computing consumption than infrastructure servers. **Our FFNet is memory and computing-friendly for the autonomous driving vehicles.** It extracts feature flow on the infrastructure side to predict future features, eliminating reliance on past data and conserving memory.  Latency compensation involves light linear computations, saving  vehicle-side computing resources.
>
> Best regards,
> 7569 Authors
>
> [1] Xu et al. OPV2V: An open benchmark dataset and fusion pipeline for perception with vehicle-to-vehicle communication. ICRA2022
> [2] Li et al. V2X-Sim: Multi-agent collaborative perception dataset and benchmark for autonomous driving. RA-L 2022
> [3] Yu et al. DAIR-V2X: A large-scale dataset for vehicle-infrastructure cooperative 3d object detection. CVPR2022
> [4] Wang et al. V2vnet: Vehicle-to-vehicle communication for joint perception and prediction. ECCV2020
> [5] Li et al. Learning distilled collaboration graph for multi-agent perception. NuerIPs2021

---

> > ### Comment · Reviewer_Wmh8 · 2023-08-21
> > **Authors have addressed most of my concerns.**
> >
> > Thanks for the answers and clarification in the rebuttal, which covered most of my concerns.

---

> > > ### Author Response · Authors · 2023-08-21
> > > **Appreciation for Your Acknowledgment**
> > >
> > > Dear Reviewer Wmh8,
> > >
> > > We are pleased that we could effectively address your concerns and we extend our gratitude for recognizing our efforts.
> > >
> > > Best Regards,
> > > 7569 Authors

---

> ### Comment · Area_Chair_H5Dt · 2023-08-18
>
> Dear Reviewer Wmh8,
>
> Please read the author's rebuttal and other reviews and indicate whether your comments have been addressed. Thank you.
>
> Best, AC

---

### Official Review · Reviewer_enaU · 2023-07-06

**Soundness:** 3 good
**Presentation:** 3 good
**Contribution:** 3 good
**Rating:** 6
**Confidence:** 5

**Summary:**

This work proposes a flow-based feature fusion framework called Feature Flow Net (FFNet) for vehicle-infrastructure cooperative 3D object detection (VIC3D).
FFNet can generate aligned features for data fusion to transmit information for fusion and solve uncertain temporal asynchrony and transmission costs.
The proposed self-supervised training of the feature flow generator lead FFNet to mitigate temporal fusion errors across various latencies.
The results on the DAIR-V2X dataset show superior performance compared to other cooperative methods while consuming only about 1/100 of the transmission cost of raw data and using a single model.

**Strengths:**

- The writing is clear and easy to follow.
- Using predicted features to solve the latency issue is neat and effective.
- The whole pipeline is simple but well-designed. It covers the transmission speed and bandwidth while keeping the model performance in the meanwhile.
- The results are solid and expressive. For different latency scenarios, FFNet achieves the best performance with less transmission data.

**Weaknesses:**

- The abbreviation of models is not easy to distinguish. For instance, FFNet versus FFNet-V2 versus FFNet-O, etc.
- The ablation of compression is missed. Without proposed compression and decompression, will it affect the tolerance of the latency of transmission?
- The last paragraph of the ablation study is not clear. The definition of the test part of the Dataset is unclear. If the models are exposed to the dataset test set, then it will be better than the one that doesn't (FFNet-O) for sure.

**Questions:**

- Although the latency and the bandwidth problem is indeed a real-world challenge, the Vehicle-Infrastructure Cooperative looks a little unrealistic. In that case, the autonomous driving vehicle will only be able to use such kind of extra information when there is an infrastructure sensor. Wouldn't it?

**Limitations:**

No.

---

> ### Author Rebuttal · Authors · 2023-08-09
>
> Dear Reviewer enaU,
> Thank you for your valuable feedback on our work. We have carefully considered your suggestions and would like to respond to each of your main comments regarding our weaknesses and questions.
>
>
> **W1.** We sincerely acknowledge your suggestion, and in our upcoming version, we will enhance the clarity of  abbreviations by incorporating a dedicated lookup table in the appendix.
>
> **W2.** Your concern about the impact of compression on latency tolerance is valid. The real-time dynamics of the road scene underscore the importance of minimizing latency for accurate predictions. As exemplified in Table 5 in our paper,  increasing latency results in a discernible accuracy loss, even with our latency compensation. Moreover, uncompressed  transmission can escalate both bandwidth consumption and latency challenges. **To illustrate, transmitting and downloading original uncompressed feature flow at 10HZ would demand a substantial 200Mb/s communication bandwidth.** This could severely impede communication responsiveness and stability, particularly in intricate traffic and communication environments. Conversely, our compression module substantially mitigates bandwidth consumption while preserving overall accuracy. The compression and decompression modules within FFNet are indeed pivotal components of our approach.
>
> | Model                         | Latency (ms) | mAP@3D |  | mAP@BEV| | Transmission cost (Bytes) |
> |-------------------------------|--------------|--------|--|-------------|-------|----|
> |                                    |                       |IoU=0.5 |IoU=0.7|      IoU=0.5 |IoU=0.7|                      |
> | FFNet                         | 200          | 55.37  | 31.66 | 63.20  | 54.69                |      1.2× 10^5    |
> | FFNet-C1                      | 200           |  55.17 |31.20 | 62.87  | 54.28                    |        **1.7×10^4 (~1/10^4)**     |
> | FFNet (without any compression) | 200           |  55.44 |31.89 | 63.97  | 55.90                    |    2.5×10^8      |
> ||||||  |   |
>
> Table 4. Evaluation results of FFNet with different compression on DAIR-V2X[1] dataset.
>
> **W3.** We wish to clarify that there are no data breach concerns. FFNet's training, including the feature flow prediction modules, is exclusively based on the training part of the DAIR-V2X[1] dataset.
>
> **Q1. Vehicle-infrastructure cooperative autonomous driving is a highly promising and dynamic field of research.** On one hand, the strategic deployment of sensors to forge intelligent transportation systems has gained attention from a diverse array of nations and research institutions, including Germany[4], China[5], and the United States[6]. Additionaly, the infrastructure sensors, including cameras, is also a common sight on city roads, highways, and various traffic networks. On the other hand, despite of achieving great progress recently, autonomous driving still faces great safety challenges due to a lack of global perspective and the limitation of long-range perception capability.  A promising solution to address these challenges is to leverage infrastructure information via Vehicle-to-Everything (V2X) communication, which has been shown to significantly expand perception range and enhance autonomous driving safety[1, 2]. Overcoming the issues in vehicle-infrastructure cooperation, thereby enabling the seamless integration of roadside information to autonomous driving vehicles, constitutes a captivating realm of research.
>
> **Furthermore, our work can be extended to more diverse V2X scenarios.** Our experiment results on the DAIR-V2X[1] and OPV2V[3] datasets confirm FFNet's efficacy across different datasets and its remarkable effectiveness in a wide range of V2X applications, including vehicle-infrastructure interactions and complex multiple vehicle scenarios. In essence, our approach facilitates autonomous driving vehicles using extra information from infrastructure or other vehicle sensors through V2X communication.
>
>
> Best regards,
> 7569 Authors
>
>
> [1] Yu et al. DAIR-V2X: A large-scale dataset for vehicle-infrastructure cooperative 3d object detection. CVPR2022
> [2] Eduardo et al. Cooperative perception for 3d object detection in driving scenarios using infrastructure sensors. TITS2020
> [3] Xu et al. OPV2V: An open benchmark dataset and fusion pipeline for perception with vehicle-to-vehicle communication. ICRA2022
> [4] https://innovation-mobility.com/en/project-providentia/
> [5] https://thudair.baai.ac.cn/index
> [6] https://www.transportation.gov/tags/v2i

---

> ### Comment · Area_Chair_H5Dt · 2023-08-18
>
> Dear Reviewer enaU,
>
> Please read the author's rebuttal and other reviews and indicate whether your comments have been addressed. Thank you.
>
> Best, AC

---

### Official Review · Reviewer_NKqy · 2023-07-06

**Soundness:** 3 good
**Presentation:** 3 good
**Contribution:** 2 fair
**Rating:** 5
**Confidence:** 5

**Summary:**

This paper introduces FFNet, a flow-based feature fusion framework that incorporates a feature flow prediction module to predict future features and addresses asynchrony issues. Instead of transmitting feature maps extracted from static images, FFNet transmits feature flow by leveraging the temporal coherence of sequential infrastructure frames. To evaluate the effectiveness of FFNet, experiments are conducted on the DAIR-V2X dataset.

**Strengths:**

1. The motivation is clear and the idea is simple yet effective.

2. The overall results look good.

**Weaknesses:**

1. The major limitation of this work pertains to its narrow scope, focusing exclusively on vehicle-infrastructure cooperative 3D object detection and mainly addressing the latency issue. While the proposed idea demonstrates effectiveness within this specific application, its applicability to broader contexts (such as multiple vehicle scenarios) remains uncertain.

2. Another weakness lies in the experimental section. The comparison made with V2VNet and DiscoNet seems unfair since these models were not originally designed to address communication latency. To ensure a more comprehensive evaluation, it would be beneficial for the authors to incorporate additional modules into these methods or compare their approach to more advanced models, such as SyncNet (ECCV 2022). This would provide a more accurate assessment of the proposed method's performance in comparison to state-of-the-art techniques.

3. A minor weakness is that the literature review is not sufficient. The authors should include more closely-related works such as [1-5]


[1] Xu, R., Xia, X., Li, J., Li, H., Zhang, S., Tu, Z., Meng, Z., Xiang, H., Dong, X., Song, R. and Yu, H., 2023. V2v4real: A real-world large-scale dataset for vehicle-to-vehicle cooperative perception. In Proceedings of the IEEE/CVF Conference on Computer Vision and Pattern Recognition (pp. 13712-13722).

[2] Li, Y., Zhang, J., Ma, D., Wang, Y. and Feng, C., 2023, March. Multi-robot scene completion: Towards task-agnostic collaborative perception. In Conference on Robot Learning (pp. 2062-2072). PMLR.

[3] Xu, R., Tu, Z., Xiang, H., Shao, W., Zhou, B. and Ma, J., 2023, March. CoBEVT: Cooperative Bird’s Eye View Semantic Segmentation with Sparse Transformers. In Conference on Robot Learning (pp. 989-1000). PMLR.

[4] Li, J., Xu, R., Liu, X., Ma, J., Chi, Z., Ma, J. and Yu, H., 2023. Learning for vehicle-to-vehicle cooperative perception under lossy communication. IEEE Transactions on Intelligent Vehicles.

[5] Su, S., Li, Y., He, S., Han, S., Feng, C., Ding, C. and Miao, F., 2023. Uncertainty quantification of collaborative detection for self-driving. ICRA.

**Questions:**

Considering the major concern about the limited scope and application, the authors may consider submitting this work to a more appropriate venue.

**Limitations:**

N.A.

---

> ### Author Rebuttal · Authors · 2023-08-09
>
> Dear Reviewer NKqy,
>
> Thank you for providing valuable feedback on our work. We will address each of the limitations you have pointed out in your comments.
>
> **W1.** Regarding the potential limitations of our work concerning application scenarios, **we have extended our  experiments to encompass the OPV2V dataset[1], which exclusively  focuses on cooperative 3D detection in multiple-vehicle scenarios.** Our forthcoming version will also include experiments conducted on the V2X-Sim[2] dataset.  Notably, the comment at the top offers a concise summary of our experiment results. These results, spanning the DAIR-V2X[3] and OPV2V[1]  datasets, show the impressive efficacy of our proposed approach across diverse V2X scenarios.
>
> In addition, we not only pay attention to the latency challenge, but also the transmission cost challenge. Specifically, we propose the Feature Flow Net (FFNet), a unified framework to overcome the hurdles posed by  uncertain temporal asynchrony and communication bandwidth limitations in cooperative 3D object detection.
>
> Overall, we kindly request that you re-evaluate the scope of our work in light of the applications we have focused on and the issues we have successfully addressed.
>
> **W2.** Addressing your concern in Weakness 2, we recognize the significance of  comparing FFNet with existing latency-aware techniques like SyncNet[4]. We wish to clarify that **we have indeed analyzed the distinctions between FFNet and SyncNet[4] in our paper (see lines 109-112).  Moreover, we have conducted comparative experiments and provide comprehensive analysis in the supplementary material** (specifically, sections F "Comparison of Feature Flow Extraction on  Different Sides" and G "Relationship to Other Existing Possible Solutions" in "S1-Appendix.pdf").
>
> It is worth noting that FFNet and SyncNet[4] approach the cooperative problem from fundamentally different angles. While SyncNet[4] focuses on leveraging historical features for latency compensation, FFNet takes a  more comprehensive approach by integrating transmission and reception considerations. **In our experiments (as outlined in section F of the supplementary material), we incorporate SyncNet's compensation module for VIC3D detection**, that is utilizing historical features for feature prediction on vehicle side. The results, as showcased in Table 2 of the appendix, illustrate FFNet's superiority in overcoming latency through feature flow extraction on infrastructure side. We also list part results in Table 3 below. This outcome aligns with the inherent challenges of extracting temporal  information from compressed features, which lack the richness found in raw sequential point clouds.
>
> | Model | Latency (ms) | mAP@3D |   |    mAP@BEV | |
> | --- | --- | --- | --- | --- |--- |
> | |   | IoU=0.5 | IoU=0.7 | IoU=0.5 | IoU=0.7 |
> | SyncNet[4]  | 100   |    50.5    |  28.25   | 58.02                   | 50.03 |
> | FFNet   | 100        |       53.46   |  30.42   | **61.20 (+3.18)** | 52.44 |
> |||||
>
> Table 3. Comparison with SyncNet[4]. Refer to appendix for more detailed experimental results.
>
> Furthermore, FFNet optimizes vehicle computing resources, as feature  flow generation transpires on infrastructure devices rather than within vehicles. This presents FFNet as a more computing-efficient solution  for resource-constrained vehicle devices. In contrast, SyncNet[4] demands  increased computational resources to leverage historical per-frame features. Additionally, SyncNet[4] necessitates heightened storage due to its dependence on past received frames. SyncNet's vulnerability to frame drops impacts its execution and performance, a concern mitigated by FFNet's storage-friendliness and robustness.
>
> **W3.** Thanks to your suggestion, we will consider adding more related papers in the next version.
>
> Best regards,
> 7569 Authors
>
> [1] Xu et al. OPV2V: An open benchmark dataset and fusion pipeline for perception with vehicle-to-vehicle communication. ICRA2022
> [2] Li et al. V2X-Sim: Multi-agent collaborative perception dataset and benchmark for autonomous driving. RA-L 2022
> [3] Yu et al. DAIR-V2X: A large-scale dataset for vehicle-infrastructure cooperative 3d object detection. CVPR2022
> [4] Lei et al. Latency-aware collaborative perception. ECCV2022

---

> > ### Comment · Reviewer_NKqy · 2023-08-22
> >
> > Thank the authors for the response. My concerns were partially resolved and I upgrade my rating to borderline accept considering the authors will add the v2v setup and missing references.

---

> > > ### Author Response · Authors · 2023-08-22
> > > **Appreciation for Your Acknowledgment**
> > >
> > > Dear Reviewer NKqy,
> > >
> > > We are delighted to have successfully addressed your concerns and greatly appreciate your recognition of our dedicated efforts.
> > >
> > > Best Regards,
> > > 7569 Authors

---

> ### Comment · Area_Chair_H5Dt · 2023-08-18
>
> Dear Reviewer NKqy,
>
> Please read the author's rebuttal and other reviews and indicate whether your comments have been addressed. Thank you.
>
> Best, AC

---

### Official Review · Reviewer_CL6E · 2023-07-08

**Soundness:** 2 fair
**Presentation:** 2 fair
**Contribution:** 2 fair
**Rating:** 6
**Confidence:** 4

**Summary:**

In this paper, a cooperative detection framework, named Feature Flow Net (FFNet), is presented to address temporal asynchrony and limited communication condition challenges in vehicle-infrastructure cooperative.
Specifically, FFNet transmits feature flow to generate aligned features for data fusion, providing a unified manner to transmit valuable information for fusion while addressing the challenges of uncertain temporal asynchrony and transmission cost.
Incorporating with self-supervised learning, the proposed FFNet framework presents a performacne improvement for VIC3D object detection with less communication cost.

**Strengths:**

1. FFNet transmits feature flow to generate aligned features for data fusion. This idea is simple yet can transmit temporal asynchrony information for fusion with less transmission cost.
2. The proposed self-supervised training approach enables FFNet with feature prediction ability to mitigate temporal fusion errors across various latencies without cooperative view and labeling.
3. FFNet demonstrates superior performance compared to other cooperative methods and demonstrates robustness in terms of latency.

**Weaknesses:**

1.  For 3D object detection, map@3D proves the detector accuracy better than map@BEV. However, it can be seen from Table 1 that FFNet's performance advantage in map@3D is much smaller than that of map@BEV, and the advantage on IoU@0.7 is less than IoU@0,5. This may indicate

     a) The proposed method does not learn the changes in the height of the environment, resulting in poor accuracy in high dimensions of FFNet.

     b) The flow-based prediction strategy is not accurate enough, which limits the upper quality of FFNet detection results.

**Questions:**

1. Experiments and visualizations are expected to analyze and verify why FFNet is deficient in high-quality detection boxes.
2. Can a finer supervision method, not limited to the global similarity, improve the performance of FFNet on map@3D?

**Limitations:**

See Weakness.

---

> ### Author Rebuttal · Authors · 2023-08-08
>
> **Resolving Anomalies in Evaluation Results using the mAP@3D Metric**
>
> Dear Reviewer CL6E:
> Thank you for your insightful question and for bringing to our attention the anomalous results in the mAP@3D metric.
>
> **Phenomenon.** We acknowledge the existence of abnormal performance in mAP@3D. As evident in Table 1, "FFNet's performance advantage in mAP@3D is much smaller than that of mAP@BEV." Moreover, both FFNet and other middle fusion methods like V2VNet[1] and DiscoNet[2] achieve significantly lower mAP@3D (IoU=0.7) compared to early fusion and late fusion methods.
>
> **Explanation.** The aforementioned issues can be attributed to the assumption of a strictly parallel ground in the implementation of feature fusion methods. Specifically, in the conversion of infrastructure Bird's Eye View (BEV)  feature/feature flow into consistent local coordinate systems, we assumed that the x-y planes would remain parallel to the ground. Unfortunately, as the DAIR-V2X dataset[4] originates from  real-world capture, the driving area does not adhere to strict parallelism. Consequently, an unintended rotation component surfaces in high dimensions when transitioning from the infrastructure's local coordinate system to that of the vehicle. Regrettably, we overlooked this high component, resulting in reduced accuracy in the high dimension. On the contrary, we used real transformation matrices in the implementation of early fusion and late fusion, mitigating the impact on the high dimension and achieving much better mAP@3D (IoU=0.7) results than middle fusion methods.
>
> **Solution**. To rectify this issue, we conducted additional experiments by unifying the bottom of all detection boxes and ground truth boxes to the same height. This effectively eliminates the influence of the high component, and we re-evaluated the detection results of FFNet. The table below shows a significant improvement in mAP@3D performance for FFNet.
>
> | High Standardization | Latency (ms) | mAP@3D |   |    mAP@BEV | |
> | --- | --- | --- | --- | --- |--- |
> | |   | IoU=0.5 | IoU=0.7 | IoU=0.5 | IoU=0.7 |
> | No | 200 | 55.37 | 31.66 | 63.20 | 54.69 |
> | Yes | 200 | **62.48 (+7.11)** | **47.92 (+16.26)** | 63.20 | 54.69 |
> | No | 300 | 53.46 | 30.42 | 61.20 | 52.44 |
> | Yes | 300 | **60.39 (+6.93)** | **45.82 (+15.40)** | 61.20 | 52.44 |
> |||||||
>
> Table 2. Evaluation results with high standardization on DAIR-V2X dataset[4].
>
> **More Discussion:** mAP@BEV is widely recognized as a more relevant metric in the autonomous driving context. Since the driving scene typically involves no objects above traffic participants, focusing on the BEV (Bird's Eye View) dimension becomes critical for evaluation. Notably, related approaches like V2VNet [1], DiscoNet [2], and V2X-ViT [3] exclusively report evaluation results that prioritize the BEV dimension. This alignment in evaluation underscores the industry's consensus on the significance of BEV-based metrics for assessing autonomous driving systems.
>
> [1] Wang et al. V2vnet: Vehicle-to-vehicle communication for joint perception and prediction. ECCV2020
> [2] Li et al. Learning distilled collaboration graph for multi-agent perception. NuerIPs2021
> [3] Xu et al. V2x-vit: Vehicle-to-everything cooperative perception with vision transformer. ECCV2022
> [4] Yu et al. DAIR-V2X: A large-scale dataset for vehicle-infrastructure cooperative 3d object detection. CVPR2022

---

> > ### Comment · Reviewer_CL6E · 2023-08-21
> >
> > Thank the authors for the response and additional experiments. My concerns were resolved and I keep my my original rating.

---

> > > ### Author Response · Authors · 2023-08-21
> > >
> > > Dear Reviewer CL6E,
> > >
> > > We sincerely appreciate your recognition and support for our work.
> > >
> > > Best Regards,
> > > 7569 Authors

---

> ### Comment · Area_Chair_H5Dt · 2023-08-18
>
> Dear Reviewer CL6E,
>
> Please read the author's rebuttal and other reviews and indicate whether your comments have been addressed. Thank you.
>
> Best, AC

---

### Author Rebuttal · Authors · 2023-08-09

**Conducting FFNet on More Datasets and Driving Contexts**

In response to some reviewers' concerns regarding the sufficiency of our experiments (specifically, Reviewer Wmh8's concern about limited dataset usage and Reviewer NKqy's concern about limited application scenarios), we have taken their feedback into consideration and conducted further experiments to address these issues.

**Datasets.** For our extended experiments, we chose to utilize two widely-used simulated datasets: OPV2X[1] and V2X-Sim[2]. These datasets are renowned for their focus on cooperative 3D detection tasks in multiple vehicle scenarios, making them ideal candidates to validate the capabilities of FFNet. Due to the constraints of the rebuttal time, we were able to complete only a portion of the experiments. Consequently, in this submission, we only report experiment results for the OPV2V dataset[1]. We will provide additional experimental results on the V2X-Sim dataset[2] in the next version.


**Experiment Setting and Results.**  Each scene in OPV2V[1] involves 2-7 autonomous vehicles. To align with  FFNet's existing framework, we adopted a two-vehicle configuration. We train FFNet, as well as the feature flow modules, using the training part of OPV2V[1]. Subsequently, we evaluate FFNet with and without feature flow prediction on the test part respect, under different latencies (100ms and 200ms). Experiment results are reported in the following table. From Table 1 below, it can be seen that:
- FFNet can leverage data from other vehicles to enhance detection performance effectively, compared to PointPillars[5].
- FFNet can mitigate performance drops caused by latency, outperforming methods like V2VNet[4] across both 100ms and 200ms latency scenarios.

| Model | FusionType       | Latency (ms) | mAP@BEV  || mAP@3D ||
|----------------------------------------|--------------|--------------|----------------|-------|----|-----|
|              |      |     |        IoU=0.5        |     IoU=0.7           |IoU=0.5| IoU=0.7  |
| PointPillars[5] | non-fusion | / |   71.7 | 56.7 | 70.0 | 44.6 |
| V2VNet[4]      | middle | 100           |       79.6        |    59.9   | 76.6                      | 49.8 |
| FFNet              | middle | 100           |       **82.1 (+10.4)**       |    63.3   | 79.6 | 54.6 |
| V2VNet[4]      | middle |  200          |       71.3        |    50.2   |             65.8          | 39.7 |
| FFNet              | middle |  200          |        **80.0 (+8.3)**       |    60.4   |  77.5 | 51.8 |
|||||||

Table 1. Experiments results on OPV2V[1] dataset.

**Conclusion.** Our experiment results on the DAIR-V2X[3] and OPV2V[1] datasets confirm FFNet's efficacy across different datasets and its remarkable effectiveness in diverse V2X scenarios, including vehicle-infrastructure interactions and complex multiple vehicle scenarios. This showcases FFNet's adaptability and potential for excelling in a wide range of V2X applications, making it a powerful and reliable solution for cooperative driving tasks, reinforcing its value and significance in V2X research and application.

[1] Xu et al. OPV2V: An open benchmark dataset and fusion pipeline for perception with vehicle-to-vehicle communication. ICRA2022
[2] Li et al. V2X-Sim: Multi-agent collaborative perception dataset and benchmark for autonomous driving. RA-L 2022
[3] Yu et al. DAIR-V2X: A large-scale dataset for vehicle-infrastructure cooperative 3d object detection. CVPR2022
[4] Wang et al. V2vnet: Vehicle-to-vehicle communication for joint perception and prediction. ECCV2020
[5] Lang et al. Pointpillars: Fast encoders for object detection from point clouds. CVPR2019

---

### Decision · Program_Chairs · 2023-09-21

**Decision:**

Accept (poster)

**Comment:**

This paper proposes the Feature Flow Net, which is presented to address temporal asynchrony and limited communication condition challenges in vehicle-infrastructure cooperative detection, and it demonstrates superior performance compared to other cooperative methods and demonstrates robustness in terms of latency. All reviewers have positive comments on the technical part, and the ACs agree with the reviewers that this paper is simple, intuitive, and effective. After reading the comments and the draft, ACs recommend accepting this paper and encourage the authors to take the comments into consideration.